# A Sample Efficient Evolutionary Strategy for Reinforcement Learning

## Abstract

We present a simple, sample-efficient algorithm for introducing large but directed learning steps in reinforcement learning (RL), through the use of evolutionary operators. The methodology uses a population of RL agents training with a common experience buffer, with occasional crossovers and mutations of the agents in order to search efficiently through the policy space. Unlike prior literature on combining evolutionary search (ES) with RL, this work does not generate a distribution of agents from a common mean and covariance matrix. It also does not require the evaluation of the entire population of policies at every time step. Instead, we focus on gradient-based training throughout the life of every policy (individual), with a sparse amount of evolutionary exploration. The resulting algorithm is shown to be robust to hyperparameter variations. As a surprising corollary, we show that simply initialising and training multiple RL agents with a common memory (with no further evolutionary updates) outperforms several standard RL baselines.

## 1 Introduction

Reinforcement learning (RL) has always faced challenges with stable learning and convergence, given its reliance on a scalar reward signal and its propensity to reach local optima (Sutton et al., 1999). While tabular RL admits theoretical analysis, there are few guarantees in the case of Deep RL. In this paper, we propose a novel way of combining Evolutionary Search (ES) with standard RL that improves the probability of converging to the globally optimal policy.

### 1.1 Motivation

Researchers have attempted to improve the sample complexity and global optimality of RL through parallelisation (Mnih et al., 2016), different batching techniques for training (Schaul et al., 2015; Khadilkar & Meisheri, 2023), reward shaping (Andrychowicz et al., 2017; Strehl & Littman, 2008), and improved exploration (Badia et al., 2020). However, all of these still focus on local incremental updates to the policies, using standard gradient-based methods. This is a barrier to effective exploration of the policy space and is heavily dependent on initialisation.

More recently, some studies have sparked a renewed interest in meta-heuristics such as evolutionary methods (Michalewicz et al., 1994) for solving RL problems. The original approach encompasses methods such as genetic algorithms (Mitchell, 1998) and simulated annealing (Van Laarhoven et al., 1987), and is based on randomised search with small or large steps, with acceptance/rejection criteria. Such algorithms have recently been extended to policy optimisation with an approach known as neuro-evolution (Salimans et al., 2017; Such et al., 2017). We describe these approaches in detail in related work, along with an intriguing compromise that combines local RL improvements with global evolutionary steps. We then propose a sample efficient version of the basic idea.

## 1.2 Related work

The earliest evolutionary search (ES) strategies for RL problems are collectively known as covariance matrix adaptation (CMA-ES), introduced by Hansen & Ostermeier (1997; 2001). Intuitively, this strategy favors previously selected mutation or crossover steps as a way to direct the search, but is difficult to scale. Hansen et al. (2003) proposed a methodology to improve convergence for a larger number of parameters, as applicable to neural networks, using higher rank (roughly analogous to higher moments of functions) information from the covariance matrix. Applications of CMA-ES to reinforcement learning (RL) include theoretical studies that focus on reliable ranking among policies (Heidrich-Meisner & Igel, 2009a;b), but their implementation is through direct policy search (without gradient-based updates) (Schmidhuber & Zhao, 1999), which is limited to simple algebraically modeled environments (Neumann et al., 2011).

CMA-ES has been used in realistic environments using different simplifications. The first option is to abstract out the problem into a simpler version. Zhao et al. (2019) take this approach for traffic signal timing. Alternatively, one may combine CMA-ES with a heuristic (Prasad et al., 2020), or with classification-based optimisation (Hu et al., 2017), or with Bayesian optimization (Le Goff et al., 2020). A more fundamental approach was taken by Maheswaranathan et al. (2019) using low-dimensional representations of the covariance matrix. In all these cases, scalability and the parallel evaluation requirements are challenging. Some studies have used ES to find the optimal architecture of the RL agent (Metzen et al., 2008; Liu et al., 2021). Yet another variant is to combine ES and RL for neural architecture search (Zhang et al., 2021). With increasing compute availability, some studies have also attempted to drop back to fundamental ES approaches to solve RL problems, with these ideas being referred to as 'neuro-evolution'. Salimans et al. (2017) and Such et al. (2017) both propose the use of 'neuro-evolution' to solve RL problems, but both methods rely on detailed reparameterization and large distributed parallel evaluation of policies.

More recently, a practical version of CMA-ES based on the cross-entropy method (CEM) (Mannor et al., 2003) has been proposed. Effectively, it is a special case of CMA-ES derived by setting certain parameters to extreme values (Stulp & Sigaud, 2012). The specific version of our interest is CEM-RL by Pourchot & Sigaud (2018), which maintains a mean actor policy $\pi_\mu$ and a covariance matrix $\Sigma$ across the population of policies. In each iteration, $n$ versions of the actor policies are drawn from this distribution (see Figure 1, left). Half of the policies are directly evaluated in the environment, while the other half receive one actor-critic update step and are then evaluated. The best $n/2$ policies are then used to update $\pi_\mu$ and $\Sigma$. The drawback of this approach is that the entire population is drawn from a single distribution, which can reduce the effectiveness of exploration. A generalised asynchronous version of CEM-RL was introduced by Lee et al. (2020), but this also has similar exploration and sampling limitations.

Apart from active RL and ES combination, some studies have used ES for experience collection and RL for training. Khadka & Tumer (2018) use a fitness metric evaluated at the end of the episode, similar to the Monte-Carlo backups used in the proposed work. However, their focus is specifically on sparse reward tasks. The mechanism is to let only the ES actors interact with the environment, collecting experience. A separate (offline) RL agent learns policies based on this experience. Periodically, the RL policy replaces the weakest policy in the ES population. The drawback of this method is that the exploration as well as parallelisation is available only to ES, and the gradient based learning is limited to a single RL policy. Another method that also collects experience with a separate policy is GEP-PG (Colas et al., 2018), which uses a goal exploration process instead of standard ES.

## 1.3 Contributions and usage

In this paper, we focus on methods and environments that do not require the massively parallel architectures of typical ES methods. We are likely to encounter such constraints wherever parallel simulations are expensive or even unavailable (for example, physical environments or finite element methods). Therefore we assume that only one policy is able to interact with the environment in one episode.

We believe that the proposed algorithm (called Evolutionary Operators for Reinforcement Learning or EORL) has the following novel and useful characteristics. First, it is very simple to implement, and modifications to standard RL and ES algorithms are minimal. Second, it requires far fewer environment interactions

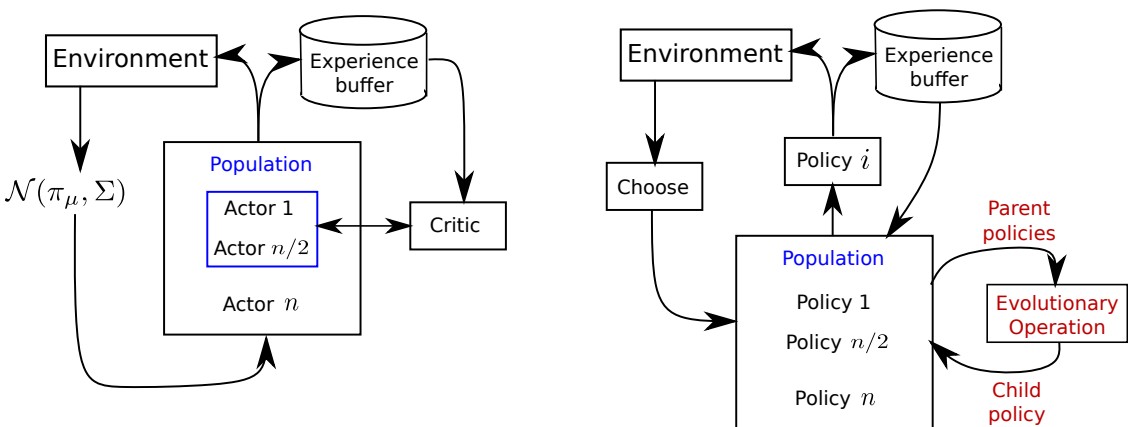

Figure 1: Comparison of the CEM-RL framework from literature (on the left) with the proposed EORL framework (on the right), using a population of $n$ policies. In CEM-RL, the whole population is generated in every time step and needs to be constantly evaluated by the environment. In EORL, every episode is run with a single policy. The common memory buffer is used to train all the individual policies. Occasionally, an evolutionary operator replaces the weakest policy in the population.

than other RL+ES approaches. Specifically, only one policy interacts with the environment at a time, thus making EORL equivalent in sample complexity to standard RL approaches[1]. Third, the existence of multiple policies allows the algorithm to have more diverse experiences than single-policy RL. Fourth, the introduction of Evolutionary Operators gives us the ability to take large search steps in the policy space, and to explore regions that are already promising.

It is important to note that EORL is *complementary* to existing off-policy RL approaches, such as ones with prioritised experience or directed exploration. Furthermore, it applies to both value based and policy gradient based methods, so long as they are off-policy.

## 2 Problem Description

We consider a standard Reinforcement Learning (RL) problem under Markov assumptions, consisting of a tuple $(\mathcal{S}, \mathcal{A}, R, P, \gamma)$, where $\mathcal{S}$ denotes the state space, $\mathcal{A}$ the set of available actions, $R$ is a real-valued set of rewards, and $P : (\mathcal{S}, \mathcal{A} \to \mathcal{S})$ is a (possibly stochastic) transition function, and $\gamma$ is a discount factor. In this paper, we limit the possible solution approaches to value-based off-policy methods (Sutton & Barto, 2018), although we shall see that the idea is extensible to other regimes as well. The chosen solution approaches focus on regressing the value of the total discounted return,

$$G_t = r_t + \gamma \, r_{t+1} + \gamma^2 \, r_{t+2} + \ldots + \gamma^{T-t} R_T,$$

where $r_t$ is the step reward at time $t$, the total episode duration is $T$, and the terminal reward is $R_T$. Further in this paper, we consider finite-time tasks with a discount factor of $\gamma = 1$, although this is purely a matter of choice and not an artefact of the proposed method. Therefore the goal of any value-based algorithm is to compute a $Q-$value approximation, $Q(s_t, a_t) \approx G_t = \sum_t^T r_t$. The standard approach for converting this approximation (typically implemented by a neural network in a form called Deep Q Networks (Mnih et al., 2015)) into a policy $\pi$, is to use an $\epsilon-$greedy exploration strategy with $\epsilon$ decaying exponentially from 1 to 0 over the course of training. Specifically,

$$\pi := \text{choose } a_t = \begin{cases} \arg\max Q(s_t, a_t) & \text{w.p. } (1 - \epsilon) \\ \text{uniform random from } \mathcal{A} & \text{w.p. } \epsilon \end{cases} \tag{1}$$

---

[1]We implicitly assume that parallelisation at least does no harm to the rate of convergence compared to standard RL approaches, while of course hoping that it actually converges even faster than standard RL

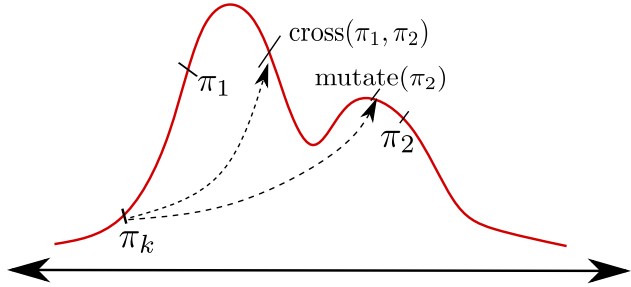

Figure 2: Intuition behind EORL. The two policies $\pi_1$ and $\pi_2$ are at good performance levels, while $\pi_k$ is performing poorly. At some random point, it can be replaced either through crossover between $\pi_1$ and $\pi_2$, or through mutation of one of the policies. This retains a high level of diversity.

Off-policy methods use a memory buffer $B$ to train the regressor $Q(s_t, a_t)$, and their differences lie in (i) the way the buffer is managed, (ii) the way memory is queried for batch training, (iii) the way rewards $r_t$ are modified for emphasising particular exploratory behaviours, and (iv) the use of multiple estimators $Q$ for stabilising the prediction. All the algorithms (proposed and baselines) in this paper follow this broad structure. In the next section, we formally define EORL, a method that utilises multiple $Q$ estimators for improving the convergence of the policy.

## 3 Methodology

### 3.1 Intuition: How and why EORL works

The intuition behind Evolutionary Operators for Reinforcement Learning (EORL) is simple, and is based on the well-established principles of parallelised random search (Price, 1977). In a high-dimensional optimization scenario, it is a well-understood phenomenon that the probability of reaching the global optimum is improved by spawning multiple random guesses and searching in their local neighbourhoods (Karp & Zhang, 1993; Zhigljavsky, 2012). EORL reuses this result by implementing parallelisation through multiple policy instances, and local search through standard gradient-based value updates. A more exhaustive search can be achieved by augmenting the local perturbations by occasional (and structured) large perturbations, in this case implemented using genetic algorithms (Mitchell, 1998). This by itself is not a novel concept, and has been used several times before.

The novelty of EORL lies in the observation that it is suboptimal to collapse the whole population into a single distribution in every time step (Pourchot & Sigaud, 2018) or to restrict the interaction of RL to offline samples (Khadka & Tumer, 2018). Instead, it is more efficient to continue local search near good solutions (see Figure 2), and to eliminate ones that are doing badly. The computational resources freed up by elimination are used to thoroughly search in promising neighborhoods. Therefore, EORL retains the gradient-based training for all but the worst policies in the current population, and even then replaces these policies only occasionally. Even the simple act of spawning multiple initial guesses, with no further evolutionary interventions, is enough to outperform most baseline algorithms.

We submit the following logical reasons for why EORL should work well.

First, EORL utilises the improved convergence characteristics of multiple random initial solutions, as described by Martí et al. (2013). The underlying theory is that of stochastic optimisation methods (Robbins & Monro, 1951), which introduced the concept of running multiple experiments to successively converge on the optimum.

Second, EORL retains the local improvement process for more promising solutions, for an extended period. This is quite important. It is well-known that the input-output relationships in neural networks are discontinuous (Szegedy et al., 2013). The $\arg\max$ selection criterion in equation 1 can lead to a significant

deviation in the state-action mapping on the basis of incremental updates to the $Q$ approximation. This cannot be predicted with single-point evaluation of policies.

Third, we know that RL has a tendency to converge to locally optimal policies (Sutton et al., 1999). This is true even of Deep RL methods because unlike supervised settings, the samples generated by online Deep RL methods are correlated. Training multiple versions with different random initializations is one workaround to this problem. Alternatively, one can use multiple policies being evaluated in parallel. However, both these approaches are very compute-heavy. EORL can be thought of as a tradeoff, where multiple randomly initialized policies are retained, but only one representative interacts with the environment at one time.

## 3.2 Evolutionary operators

The evolutionary operators that we define are described below, and are based on standard operators defined by Michalewicz et al. (1994). We assume that the policy $\pi_i$ is parameterised by $\Theta_i$, a vector composed of $P$ scalar elements (weights and biases) $\theta_{i,p}$, with $p \in \{1, \ldots, P\}$. All policies $\pi_i$ contain the same number of parameters, since they have the same architectures. In the following description, we represent the newly spawned child policy by $\pi_c$, and the parent policies by $\pi_i$ and $\pi_j$ (the latter only when applicable). The 'fitness' of $\pi_i$ is given by a scalar value $A_i$, further elaborated later.

*O-1* Random crossover: The cross ratio is given by $(\tau, 1 - \tau) = \text{softmax}(A_i, A_j)$. Then every parameter of $\pi_c$ is chosen with probability $\tau$ from $\pi_i$ with a multiplicative noise factor. Specifically,

$$\theta_{c,p} = \begin{cases} \theta_{i,p} \cdot \mathcal{N}(1, \sigma) & \text{w.p. } \tau \\ \theta_{j,p} \cdot \mathcal{N}(1, \sigma) & \text{w.p. } (1 - \tau) \end{cases},$$

where $\mathcal{N}(1, \sigma)$ is a random variable drawn from a normal distribution with mean 1. The multiplicative noise scales parameters proportional to their magnitude, but importantly it does not change the expectation of the product. Since intuitively neural networks are multiplicative chains of weights, perturbing with multiplicative noise ensures that the signal is not unnecessarily dimmed. In this paper, we use $\sigma = 0.25$.

*O-2* Linear crossover: We again work with a cross ratio of $(\tau, 1 - \tau) = \text{softmax}(A_i, A_j)$, but now the $\tau$ simply defines the weight for averaging of the parameters:

$$\theta_{c,p} = (\tau \, \theta_{i,p} + (1 - \tau) \, \theta_{j,p}) \cdot \mathcal{N}(1, \sigma)$$

*O-3* Random mutation: This operator only has a single parent policy $\pi_i$, and generates $\pi_c$ solely with multiplicative noise:

$$\theta_{c,p} = \theta_{i,p} \cdot \mathcal{N}(1, \sigma)$$

## 3.3 Specification

The formal definition of EORL is given in Algorithm 1, which is best understood after we define the following terms. Consider a scenario where there is a fixed population size of $n$ policies $\pi_i$, $i \in \{1, \ldots, n\}$ throughout the course of training. The fitness $A_i$ of policy $\pi_i$ undergoes a soft update after every episode that is run according to $\pi_i$. Specifically,

$$A_i \leftarrow q \, A_i + (1 - q) \sum_{t=1}^{T} r_t, \tag{2}$$

for an episode of $T$ steps that runs using $\pi_i$ and with a user-defined weight $q$. All $A_i$ are initialised to 0 at the start of training. If a child policy $\pi_c$ is generated using an evolutionary operator (Section 3.2) from policies $\pi_i$ and $\pi_j$ in ratio $\tau : (1 - \tau)$, then the fitness is reset according to $A_c \leftarrow \tau A_i + (1 - \tau) A_j$. Note that the update equation 2 is carried out only for one policy per episode. Furthermore, using a large value of $q$ (we use $q = 0.9$) keeps the estimates stable in stochastic environments.

The choice of policy in every episode is made using $\epsilon-$greedy principles. With a probability $\epsilon$, a policy is chosen randomly among the $n$ available policies. With a probability $(1 - \epsilon)$, we choose the best-performing

---

**Algorithm 1** Implementation of Evolutionary Operators for RL (EORL)

---

1: Define: Population size $n$, crossover and mutation rates $\kappa$ and $\mu$
2: Initialise: Buffer $B \leftarrow \emptyset$, fitness values $A_i \leftarrow 0$, $i \in \{1, \ldots, n\}$
3: **for** episode $e \in \{1, \ldots, E\}$ **do**           ▷ Outer coordination loop
4:   Reset environment
5:   Choose agent policy $\pi_e$ from $\{\pi_1, \ldots, \pi_n\}$       ▷ See section 3.3
6:   Run episode using $\pi_e$ until timeout or goal reached
7:   Add samples to buffer $B$
8:   Train $\pi_i$, $i \in \{1, \ldots, n\}$ with independently sampled mini-batches from $B$
9:   **if** evolutionary operation is called for **then**
10:    Choose operator from *O-1*, *O-2*, *O-3*       ▷ See section 3.2
11:    Pick parent policy/policies from $\pi_i$, $i \in \{1, \ldots, n\}$ from top-50 percentile
12:    Pick child policy $\pi_k$, $k = \arg\min(A_i)$ to be replaced
13:    Replace $\pi_k$ with newly generated policy
14:    Set $A_k \leftarrow$ weighted average of parent policy/policies
15:   **end if**
16: **end for**

---

policy among the group, based on their values of $A_i$. If there are multiple policies at the highest level of $A_i$, a policy is chosen randomly among the best-performing subgroup. The only exception to this rule is when a newly generated (through evolutionary operators) policy exists; in this case the newly generated policy is chosen to run in the next episode. At the end of every episode, an evolutionary operator may be called according to one of the following two schemes:

**Uniform Random:** We define fixed values of crossover rate $\kappa$ and mutation rate $\mu$. At the end of every episode $e$ out of a total training run of $E$ episodes, a crossover operation may be called with a probability $\kappa(1 - e/E)$, decaying linearly over the course of training. If called, one of the two crossover operators (*O-1* and *O-2*) is chosen with equal probability. If the crossover operation is not called, the mutation operator *O-3* is called with probability $\mu(1 - e/E)$. In both cases, the parent policy/policies is/are chosen randomly from the top 50 percentile of policies.

**Active Random:** An obvious alternative to the predefined annealing schedule as described above, is an active or dynamic probability of calling the evolutionary operators. We first define a 'reset' point $e^*$, which is the episode when either (i) an evolutionary operator was last called, or (ii) a total reward in excess of 95% of the best observed reward was last collected. The policy selection is identical to the uniform random method (above) until the exploration rate $\epsilon$ decays to 0.05. At this point, the multiplier for rates $\kappa$ and $\mu$ switches from $(1 - e/E)$ to $[(e - e^*)/n]$, clipped between $(1 - e/E)$ and 5. Essentially, we linearly *increase* the probability of an evolutionary operation if good rewards are not being consistently collected towards the end of training.

## 4 Results

### 4.1 Baseline algorithms

The results presented in this paper compare 5 versions of EORL with 6 baseline value-based off-policy algorithms. Among EORL versions, three versions use the Uniform Random evolutionary option, with $(\kappa, \mu)$ given by $(0.05, 0)$, $(0.05, 0.05)$, and $(0.1, 0.05)$ respectively. They are referred to respectively as EORL-05-00, EORL-05-05, and EORL-10-05 in results. The fourth version of EORL (EORL-ACTV) uses the Active Random procedure. Finally, we run a baseline without evolutionary operations but with $n$ initialised policies (EORL-FIX) as an ablation study of the effect of crossovers and mutations. This is effectively EORL with $\kappa = \mu = 0$.

All the 11 algorithms use identical architectures as described below, with learning rates of 0.01, a memory buffer size of 100 times the timeout value of the environment, a training batch size of 4096 samples, and

training for 2 epochs at the end of each episode. All 10 random seeds for each algorithm and environment version are run in parallel on a $10-$core CPU with 64 GB RAM. The number of policies for CEM-RL and the various versions of EORL is set to $n = 8$ based on ablation experiments described later.

We briefly describe the remaining baselines here. More details are given in Appendix A. Vanilla DQN (called VAN) (Mnih et al., 2015) and Contrastive Experience Replay (CER) (Khadilkar & Meisheri, 2023) are run with standard hyperparameter settings as described by the authors (apart from the ones defined above). Prioritised Experience Replay (PER) (Schaul et al., 2015) is run with $\beta$ increasing linearly from 0.4 to 1.0 during training, and $\alpha = 0.6$. Hindsight Experience Replay (HER) (Andrychowicz et al., 2017) uses a goal buffer size of 8 to match the number of parallel policies in EORL, containing the terminal states of the best 8 episodes during training. Count-based exploration (CBE) (Strehl & Littman, 2008) augments the true step reward provided by the environment with a count-based term $\beta/\sqrt{1 + N(s, a)}$, where $N$ is the number of visits to the particular state-action pair $(s, a)$. We use $\beta = 1$ and un-normalised $N$ after a reasonable amount of fine-tuning.

Cross-Entropy Method (CEM-RL) was introduced by Pourchot & Sigaud (2018). We have to modify the basic procedure in order to retain comparable sample complexity. Instead of parallel evaluation of all $n$ policies, we run them in round-robin fashion to minimise update latency. If the outcome is in the bottom 50 percentile, the policy is replaced (with probability 0.5) by another drawn from the best existing policies. Otherwise, it is put in the pool of RL policies. All policies in the RL pool (roughly 3/4 of the population) are trained using gradient-based updates after every episode.

Note that we do not compare EORL with methods such as Rainbow (Hessel et al., 2018), because we wish to observe the individual contributions of the modifications given above. Since EORL is a complementary (rather than competing) approach to each of the above, it can always be used as one more tool in addition to the ensemble included in Rainbow.

## 4.2 Environments

We select the following two simple environments for demonstrating the results. We also emphasise that EORL is designed to work in conjunction with other off-policy algorithms, and so the point of experimentation is to emphasise its consistency of performance. Experiments on more complex environments are deferred to future work because of a lack of timely access to the required compute infrastructure.

### 4.2.1 1D bit-flipping

The first task is adapted from Hindsight Experience Replay (Andrychowicz et al., 2017). We consider a binary number of $m$ bits, with all zeros being the starting state, and all ones as the goal state. The agent observes the current number as the $m$-dimensional state. The action set of the agent is a one-hot vector of size $m$, indicating which bit the agent wants to flip. The step reward is a constant value of $\frac{-1}{5m}$ for every bit flip that does not result in the goal state, and a terminal reward of $+10$ for arriving at the goal state. There is a timeout of $5m$ moves. In a modified version of the environment, we introduce a '**subgoal**' state consisting of alternating zeros and ones. The agent gets a reward of $+10$ if it gets to the goal after passing through the subgoal, and only $+1$ if it goes directly to the goal. This modification increases the exploration complexity of the environment.

All the algorithms in this version of the environment are trained on 10 random seeds, 400 episodes per seed, and an $\epsilon-$greedy decay multiplier of 0.99 per episode. All the agents (except HER) use a fully connected network with an input of size $m$, two hidden layers of size 32 and 8 respectively with `ReLU` activation, and a linear output layer of size $m$. Since HER augments the input state with the intended goal, its input size is $2m$. Rewards are computed using Monte-Carlo style backups.

### 4.2.2 2D grid navigation with subgoals

The second environment is adapted from Contrastive Experience Replay (Khadilkar & Meisheri, 2023). We consider an $m \times m$ grid, with the agent spawning at position $[1, 1]$ in each episode, and the goal state at position $[m, m]$ as shown in Figure 3. There are 4 possible actions in each step, `UP,DOWN,LEFT,RIGHT`.

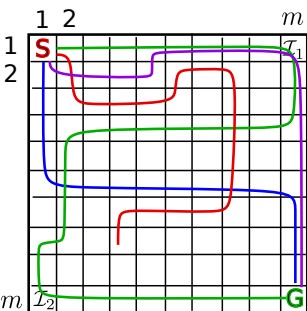

Figure 3: A graphical illustration of the 2D grid environment, showing two subgoals. The significance of different trajectories is explained in text.

Multiple versions of the same task can be produced by varying the effect of the subgoals $\mathcal{I}_1$ and $\mathcal{I}_2$ located at $[1, m]$ and $[m, 1]$ respectively.

**Subgoal 0**: In the most basic version, visiting either or both subgoals on the way to the goal has no effect. If $tmax$ is the timeout defined for the task, the agent gets a reward of $-1/tmax$ for every step that does not lead to a goal, and a terminal reward of $+10$ for reaching the goal.

**Subgoal 1**: The exploration can be made more challenging by 'activating' one subgoal, $\mathcal{I}_1$. The agent gets a terminal reward of $+10$ if it visits subgoal $\mathcal{I}_1$ at position $[1, m]$ on the way to the goal, and a smaller terminal reward of $+1$ for going to the goal directly without visiting $\mathcal{I}_1$. The other subgoal $\mathcal{I}_2$ has no effect. Among the trajectories shown in Figure 3, the green and purple ones will get a terminal reward of $+10$, the blue one gets $+1$, and the red one gets 0.

**Subgoal 2+**: In this case, both subgoals are activated. The agent gets a reward of $+10$ only if both $\mathcal{I}_1$ and $\mathcal{I}_2$ are visited on the way to the goal (regardless of order between them). A terminal reward of $+2$ is given for visiting only one subgoal on the way, and a reward of $+1$ is given for reaching the goal without visiting any subgoals (blue= $+1$, purple= $+2$, green= $+10$, red= 0 in Figure 3).

**Subgoal 2-**: A final level of complexity is introduced by providing a negative reward for visiting only one subgoal. The agent gets a reward of $+10$ only if both $\mathcal{I}_1$ and $\mathcal{I}_2$ are visited on the way to the goal (regardless of order between them). A terminal reward of $-1$ is given for visiting only one subgoal on the way, and a reward of $+1$ is given for reaching the goal without visiting any subgoals.

All the algorithms in this version of the environment are trained on 10 random seeds, but with varying episode counts and decay rates based on complexity of the task. The timeout is set to 10 times the length of the optimal path. All the agents (except HER) use a fully connected feedforward network with an input of size 4 (consisting of normalised $x$ and $y$ position and binary flags indicating whether $\mathcal{I}_1$ and $\mathcal{I}_2$ respectively have been visited in the current episode), two hidden layers of size 32 and 8 respectively with `ReLU` activation, and a linear output layer of size 4. Since HER augments the input state with the intended goal, its input size is 8. Rewards are computed using Monte-Carlo backups.

**Introducing stochasticity:** Finally, we create stochastic versions of each of the 2D environments by introducing different levels of randomness in the actions. With a user-defined probability, the effect of any action in any time step is randomised uniformly among the movement directions.

### 4.3 Experimental results

The saturation rewards (averaged over the last 100 training episodes) achieved by all algorithms on the 1D environment are summarised in Table 1, with size $m$ between 6 and 10. Among the algorithms, the Active Random version of EORL has the highest average reward, as well as the greatest number of instances (four) with the highest reward. All the algorithms fail to learn for the environment with $m = 10$ and 1 subgoal.

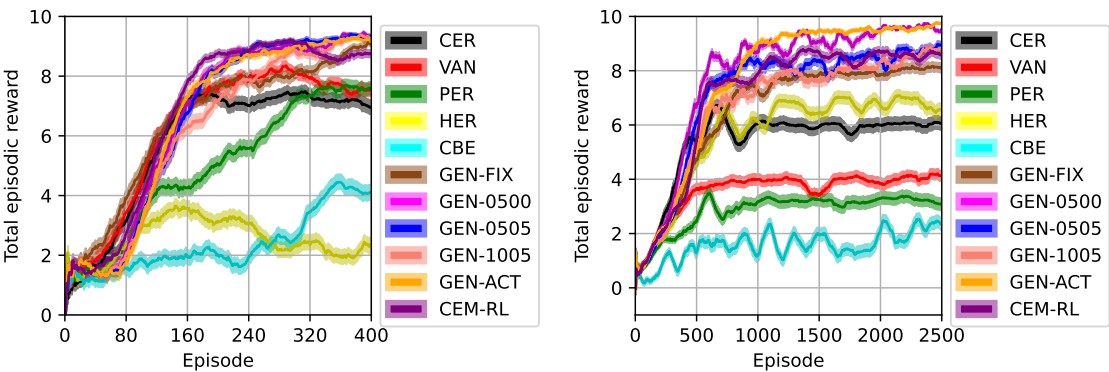

Figure 4: Sample training plots, from [left] the 1D environment of size 6 without subgoals, and [right] the 2D subgoal 2+ environment, size 8×8. Each algorithm is run for 10 random seeds, with shaded regions denoting the standard deviation across random seeds. Full results in appendix.

A full set of training plots is included in Appendix C. Note also that HER results deviate from the original paper Andrychowicz et al. (2017) because of the much shorter episode length, as well as the subgoal.

Along similar lines, Table 2 summarizes the results for the 2D navigation environment. There are many variations possible in this case, in terms of subgoals, stochasticity, and grid size. Hence the full table (with 56 instances) is provided in Appendix B. The summary table also shows interesting characteristics, with a Uniform Random version ($\kappa = 0.05, \mu = 0.0$) outperforming the other environments in terms of average reward as well as the number of times it is the best-performing algorithm. Although EORL-ACTV is fourth in terms of average reward, it is ranked second in terms of best-results. Figure 4 shows a sample training plot.

We make the following observations from a general analysis of the results. **First**, that the EORL algorithms are dominant across the whole range of experiments, being the best-performing ones in 60 of 66 instances across the two environments (with one instance – 1D, size 10, 1 subgoal – having no winners). This is consistent with the ranking according to average rewards as well. **Second**, we note that the milder versions of EORL (lower $\kappa$ and $\mu$) perform better for simpler environments, either with smaller size or with fewer subgoals. We may conclude that harder environments need higher evolutionary churn, which is intuitive. **Third** and most surprising, we see that EORL-FIX with $n$ policies and no evolutionary operations consistently outperforms all baselines apart from CEM-RL. This is a strong indication that the idea of running multiple random initialisations within a single training has merit.

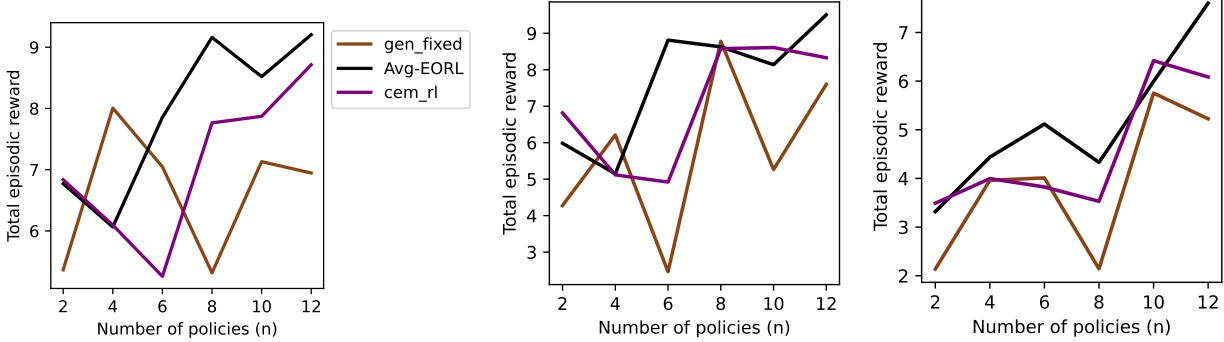

Figure 5: Effect of number $n$ of parallel policies being maintained by various algorithms. All the four EORL versions are averaged for clarity, and the plots correspond to the average total reward in the last 100 episodes of training for 10 random seeds. We consider 2D grid with size 16×16, 20×20, and 40×40 respectively, all with 1 subgoal, trained for 1000 episodes at an $\epsilon$ decay rate of 0.995.

Table 1: 1D bit-flipping environment results with a decay rate of 0.99 and 400 episodes, averaged over 10 random seeds. Reported values are average total reward in the last 100 episodes of training. The last row indicates the number of experiments in which a particular algorithm was the best-performing.

| Size | Sub goal | VAN | CER | HER | PER | CBE | EORL FIX | CEM RL | EORL 05-00 | EORL 05-05 | EORL 10-05 | EORL ACTV |
|---|---|---|---|---|---|---|---|---|---|---|---|---|
| 6 | 0 | 7.69 | 7.07 | 2.19 | 7.59 | 4.21 | 8.80 | 8.68 | 9.19 | **9.29** | 9.12 | 9.14 |
| 7 | 0 | 8.11 | 6.90 | −0.77 | 5.62 | 0.11 | 7.06 | 8.20 | 8.90 | 8.22 | 8.52 | **8.86** |
| 8 | 0 | 4.78 | 2.78 | −0.84 | 5.14 | −0.95 | 4.02 | 3.90 | **6.05** | 4.78 | 5.04 | 4.49 |
| 9 | 0 | 0.79 | −0.86 | −1.00 | −0.71 | −1.00 | −0.93 | −0.27 | 0.01 | 1.03 | −0.92 | **1.81** |
| 10 | 0 | −1.00 | −1.00 | −1.00 | −0.97 | −1.00 | −1.00 | −1.00 | −1.00 | −1.00 | −1.00 | **−0.94** |
| 6 | 1 | 6.94 | 9.03 | 0.33 | 7.14 | 0.01 | 8.89 | **9.12** | 8.27 | 6.34 | 7.02 | 8.81 |
| 7 | 1 | 0.27 | 1.66 | −1.00 | 0.96 | −1.00 | 1.73 | 1.77 | 1.88 | **2.80** | 1.38 | 1.86 |
| 8 | 1 | −0.58 | −0.27 | −1.00 | −0.30 | −1.00 | −0.67 | −0.52 | −0.41 | −0.73 | **0.43** | −0.67 |
| 9 | 1 | −0.99 | −0.89 | −1.00 | −1.00 | −1.00 | −0.86 | −1.00 | −0.86 | −0.86 | −1.00 | **−0.78** |
| 10 | 1 | −1.00 | −1.00 | −1.00 | −1.00 | −1.00 | −1.00 | −1.00 | −1.00 | −1.00 | −1.00 | −1.00 |
| Average | | 2.50 | 2.34 | −0.51 | 2.25 | −0.26 | 2.60 | 2.79 | 3.10 | 2.89 | 2.76 | **3.16** |
| Best results | | 0 | 0 | 0 | 0 | 0 | 0 | 1 | 1 | 2 | 1 | 4 |

Table 2: Summary of 2D grid results, aggregated over environment sizes for various subgoal versions and stochasticity levels. There are a total of 56 variations. A full list of results is provided in Appendix B. The environment sizes range from 8×8 to 80×80. Reported values are average reward in the last 100 episodes of training. The last row indicates the number of experiments in which a particular algorithm was the best-performing one (counted as 0.5 in case of ties).

| Sub goal | Stoch | VAN | CER | HER | PER | CBE | EORL FIX | CEM RL | EORL 05-00 | EORL 05-05 | EORL 10-05 | EORL ACTV |
|---|---|---|---|---|---|---|---|---|---|---|---|---|
| 0 | 0.00 | 6.59 | 6.89 | 6.55 | 6.35 | 3.59 | 8.03 | 8.74 | **9.34** | 9.14 | 9.55 | 9.17 |
| 0 | 0.10 | 7.10 | 6.51 | 6.86 | 6.31 | 4.22 | 8.89 | 8.89 | 9.63 | 9.52 | **9.65** | 9.43 |
| 0 | 0.20 | 7.41 | 7.60 | 6.54 | 6.56 | 4.53 | 8.73 | 8.51 | **9.79** | 9.62 | 9.59 | 9.69 |
| 1 | 0.00 | 3.30 | 3.75 | 4.15 | 2.90 | 1.00 | 5.40 | 6.14 | **7.56** | 7.27 | 7.14 | 6.56 |
| 1 | 0.10 | 3.27 | 3.48 | 3.91 | 3.53 | 0.53 | 5.60 | 6.62 | **7.35** | 7.23 | 7.31 | 6.43 |
| 1 | 0.20 | 4.01 | 4.10 | 4.01 | 3.62 | 0.78 | 5.76 | 5.81 | **6.92** | 6.73 | 6.75 | 6.20 |
| 2+ | 0.00 | 1.87 | 2.36 | 1.75 | 1.33 | −0.19 | 2.69 | 2.75 | 3.16 | 3.08 | 2.99 | **3.34** |
| 2+ | 0.10 | 1.99 | 3.05 | 2.81 | 1.91 | 0.37 | 3.15 | 3.58 | 3.78 | 3.34 | **3.83** | 3.50 |
| 2- | 0.00 | 1.75 | 1.94 | 1.29 | 1.08 | 0.10 | 2.09 | 2.53 | 3.15 | 3.14 | **3.53** | 3.12 |
| Average | | 4.43 | 4.65 | 4.48 | 4.02 | 1.84 | 5.96 | 6.32 | **7.16** | 6.98 | 7.10 | 6.77 |
| Best results | | 1 | 3 | 0 | 0 | 0 | 6 | 0 | 19.5 | 9 | 8 | 9.5 |

### 4.4 Ablation to understand the effect of $n$

Apart from the crossover and mutation rates (of which we have already shown results with multiple combinations), the only other design decision unique to EORL is the value of $n$, the size of the policy population. Figure 5 shows the average reward in the last 100 episodes of training for 10 random seeds, for various algorithms. The three Uniform Random and the Active Random version of EORL are averaged into a single plot for visual clarity. We can see that all algorithms show a roughly increasing trend as $n$ increases from 2 to 12, with the relatively smaller environments (16×16 and 20×20) showing some signs of saturation for $n \geq 8$. This is why we choose $n = 8$ for the bulk of our experiments, as a compromise between the computational/memory intensity and the performance level.

## 5 Conclusion and limitations

We showed that there is a simple way of introducing evolutionary ideas into reinforcement learning, without increasing the number of interactions required with the environment.

One obvious limitation of the proposed approach is its difficulty in adaptation to on-policy algorithms. Specifically, the basis of EORL is to use samples collected by other policies for training any given policy in the population. This directly violates the on-policy assumption. However, it is possible that some form of policy similarity metric may be used to bridge this gap. This is a thread for future work.

Additionally, a more thorough investigation of the present version of EORL on a variety of environments is under way, including those with continuous state-action spaces.

**Broader Impact Statement**

This work does not have any ethical implications or risks as far as the authors can foresee.

**Acknowledgments**

Withheld for purposes of anonymity.

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

## A  Details of baseline implementations

1. Vanilla DQN as described by Mnih et al. (2015), shortened to VAN in results.

2. Count-based exploration (CBE) as introduced by Strehl & Littman (2008). As per their MBIE-EB model, we augment the true step reward provided by the environment with a count-based term $\beta/\sqrt{1 + N(s, a)}$, where $N$ is the number of visits to the particular state-action pair $(s, a)$. We use $\beta = 1$ and un-normalised $N$ after a reasonable amount of fine-tuning. Higher values of $\beta$ led to too much exploration considering the number of training episodes, while lower values of $\beta$ behaved identically to vanilla DQN.

3. Hindsight Experience Replay (HER), introduced by Andrychowicz et al. (2017). We use a goal buffer size of 8 to match the number of parallel policies in EORL, containing the terminal states of the best 8 episodes during training. HER is the only algorithm among the 11 which has a different input size, in order to accommodate the goal target in addition to the state. The specifics are given in Section 4.2.

4. Prioritised Experience Replay (PER), introduced by Schaul et al. (2015). We use the hyperparameter values as reported by Khadilkar & Meisheri (2023), with $\beta$ increasing linearly from 0.4 to 1.0 during training, and $\alpha = 0.6$.

5. Contrastive Experience Replay (CER), proposed by Khadilkar & Meisheri (2023). Since the second environment used in this paper is adapted from CER, we use the same hyperparameter settings reported by the authors.

6. Cross-Entropy Method (CEM-RL), introduced by Pourchot & Sigaud (2018). For a fair comparison with the other algorithms, we cannot spawn and evaluate the entire population of policies in every episode (as originally proposed). Instead, we evaluate policies by choosing them in a round-robin manner for every episode. If the outcome is in the bottom 50 percentile, the policy is replaced (with probability 0.5) by another drawn from the best existing policies. Otherwise, it is put in the pool of RL policies. All policies in the RL pool (roughly 3/4 of the population) are trained using gradient-based updates after every episode.

7. Fixed population of policies (EORL-FIX). This is a basic version of EORL which spawns $n$ policies at the start and trains them throughout using gradient-based RL, with no evolutionary operations. Effectively, this is EORL with $\kappa = \mu = 0$.

For further details, please refer to the attached codes and readme file.

## B   Detailed tabulated results for all environments

Table 3: Full set of results on 2D navigation environment, averaged over 10 random seeds. Reported values are average total reward in the last 100 episodes of training. Version of environment is given by the subgoals column, and size indicates one side of the square (for example, size of 80 implies an $80{\times}80$ sized environment). The number of episodes run for training each seed are given in the $E$ column. Note that $E = 1000$ was run with an $\epsilon$ decay rate of 0.995 per episode, $E = 2500$ was run with a decay rate of 0.998 per episode, and $E = 4000$ with a decay rate of 0.999 per episode. The stochasticity (probability of choosing a random action) is also listed. The last two rows provide aggregate results, with average total reward and the number of times a given algorithm had the best performance in an experiment (counted as 0.5 for every tie between two algorithms).

| Size | Sub goal | Stoch | $E$ | VAN | CER | HER | PER | CBE | EORL FIX | CEM RL | EORL 05-00 | EORL 05-05 | EORL 10-05 | EORL ACTV |
|------|------|-------|-----|-----|-----|-----|-----|-----|----------|--------|------------|------------|------------|-----------|
| 8  | 0 | 0.00 | 1000 | 7.64 | 6.90 | 9.63 | 7.70 | 7.93 | 8.58 | 9.65 | 9.58 | **9.75** | 9.71 | 9.71 |
| 12 | 0 | 0.00 | 1000 | 6.42 | 7.57 | 7.63 | 7.34 | 7.18 | 7.69 | 9.51 | **9.81** | 9.70 | 9.75 | 9.80 |
| 16 | 0 | 0.00 | 1000 | 8.71 | 8.81 | 9.38 | 8.75 | 6.46 | **9.88** | 9.59 | 9.78 | 9.86 | 9.71 | 9.74 |
| 20 | 0 | 0.00 | 1000 | 9.80 | **9.83** | 8.63 | 8.76 | 1.14 | 8.79 | 9.37 | 9.60 | 8.65 | 9.74 | 9.77 |
| 40 | 0 | 0.00 | 1000 | 6.40 | 5.44 | 3.12 | 5.54 | 0.92 | 8.80 | 9.46 | 9.35 | 9.69 | 9.40 | **9.84** |
| 60 | 0 | 0.00 | 1000 | 4.89 | 7.49 | 5.45 | 3.03 | 1.42 | 8.76 | 8.19 | **9.62** | 9.38 | 9.44 | 8.53 |
| 80 | 0 | 0.00 | 1000 | 2.29 | 2.21 | 2.02 | 3.35 | 0.09 | 3.69 | 5.40 | 7.66 | 6.94 | **9.06** | 6.78 |
| 8  | 0 | 0.10 | 1000 | 7.03 | 7.39 | 7.73 | 8.79 | 9.30 | 8.73 | 9.76 | 9.71 | 9.81 | 9.87 | **9.88** |
| 12 | 0 | 0.10 | 1000 | 8.60 | 8.29 | 9.74 | 7.54 | 8.13 | 8.78 | 9.68 | **9.86** | 9.82 | 9.81 | **9.86** |
| 16 | 0 | 0.10 | 1000 | 7.70 | 6.65 | 9.76 | 8.63 | 7.17 | 9.84 | 9.57 | 9.78 | **9.86** | 9.79 | 9.85 |
| 20 | 0 | 0.10 | 1000 | 8.40 | 8.69 | 7.27 | 9.76 | 3.57 | **9.89** | 9.73 | 9.87 | 9.81 | 9.84 | 9.86 |
| 40 | 0 | 0.10 | 1000 | 8.77 | 7.61 | 4.74 | 6.37 | 0.07 | 8.70 | 9.41 | **9.85** | 9.67 | 9.45 | 9.84 |
| 60 | 0 | 0.10 | 1000 | 6.22 | 5.34 | 7.16 | 1.93 | 2.10 | 8.76 | 7.72 | **9.82** | 9.15 | 9.56 | 9.44 |
| 80 | 0 | 0.10 | 1000 | 2.99 | 1.59 | 1.63 | 1.16 | $-0.82$ | 7.55 | 6.36 | 8.51 | 8.55 | **9.26** | 7.31 |
| 8  | 0 | 0.20 | 1000 | 9.12 | 9.86 | 9.86 | 9.01 | 9.69 | 9.86 | 9.71 | **9.87** | 9.86 | 9.85 | **9.87** |
| 12 | 0 | 0.20 | 1000 | 9.47 | 9.26 | 8.75 | 9.60 | 7.35 | **9.87** | 9.70 | 9.76 | 9.82 | 9.79 | 9.77 |
| 16 | 0 | 0.20 | 1000 | 9.60 | 9.83 | 8.80 | 7.73 | 6.88 | 8.78 | 9.53 | 9.80 | **9.86** | 9.83 | 9.82 |
| 20 | 0 | 0.20 | 1000 | 9.16 | 8.15 | 7.47 | 8.79 | 5.02 | **9.87** | 9.69 | 9.86 | 9.79 | 9.83 | 9.84 |
| 40 | 0 | 0.20 | 1000 | 4.07 | 5.52 | 5.53 | 4.42 | 1.52 | 8.73 | 8.54 | 9.84 | 9.77 | 9.67 | **9.87** |
| 60 | 0 | 0.20 | 1000 | 4.42 | 4.41 | 4.19 | 3.10 | 1.06 | 7.48 | 7.99 | 9.74 | **9.78** | 9.32 | **9.78** |
| 80 | 0 | 0.20 | 1000 | 6.03 | 6.19 | 1.20 | 3.28 | 0.19 | 6.55 | 4.39 | **9.67** | 8.49 | 8.87 | 8.89 |

| Size | Sub goal | Stoch | $E$ | VAN | CER | HER | PER | CBE | EORL FIX | CEM RL | EORL 05-00 | EORL 05-05 | EORL 10-05 | EORL ACTV |
|------|------|------|------|------|------|------|------|------|------|------|------|------|------|------|
| 8 | 1 | 0.00 | 1000 | 6.59 | 7.81 | 9.58 | 6.15 | 6.44 | 8.61 | 9.47 | **9.81** | 9.70 | 9.80 | 9.73 |
| 12 | 1 | 0.00 | 1000 | 5.80 | 6.89 | 6.58 | 7.76 | 4.85 | 8.69 | 8.71 | 9.66 | 8.74 | 7.83 | **9.69** |
| 16 | 1 | 0.00 | 1000 | 5.81 | 3.57 | 6.41 | 4.87 | −0.75 | 5.36 | 6.94 | **9.79** | 9.60 | 9.59 | 7.97 |
| 20 | 1 | 0.00 | 1000 | 2.29 | 3.34 | 3.13 | 2.21 | −0.99 | 8.67 | 9.44 | 7.79 | **9.59** | 7.91 | 6.15 |
| 40 | 1 | 0.00 | 1000 | 0.91 | 1.39 | 2.86 | 0.77 | −0.82 | 2.14 | 3.53 | 4.11 | **4.90** | 4.08 | 4.22 |
| 40 | 1 | 0.10 | 1000 | 2.01 | 1.38 | 0.47 | 2.03 | −0.87 | 5.97 | 5.29 | 5.52 | 4.14 | **6.76** | 4.56 |
| 60 | 1 | 0.00 | 1000 | 1.14 | 1.00 | 1.37 | −1.00 | −0.92 | 1.15 | 2.32 | 6.46 | 3.88 | **6.55** | 2.02 |
| 80 | 1 | 0.00 | 1000 | 0.59 | 2.29 | −0.91 | −0.47 | −0.79 | 3.19 | 2.56 | 5.32 | 4.47 | 4.23 | **6.10** |
| 8 | 1 | 0.10 | 1000 | 7.68 | 6.56 | 8.73 | 7.55 | 4.96 | 7.65 | 9.78 | **9.83** | 9.76 | 9.69 | 9.77 |
| 12 | 1 | 0.10 | 1000 | 5.08 | 7.93 | 8.48 | 6.97 | 1.36 | 7.14 | 9.43 | 8.85 | 9.25 | **9.55** | 8.91 |
| 16 | 1 | 0.10 | 1000 | 3.85 | 2.40 | 6.79 | 6.07 | 0.91 | 7.90 | 8.29 | **9.85** | 7.91 | 8.62 | 7.81 |
| 20 | 1 | 0.10 | 1000 | 4.50 | 3.91 | 2.94 | 2.33 | −0.87 | 5.33 | 8.42 | **8.90** | 8.75 | 7.66 | 5.92 |
| 60 | 1 | 0.10 | 1000 | 0.07 | 1.91 | −0.47 | 0.26 | −0.76 | 3.65 | 3.62 | 4.45 | 4.24 | **5.72** | 3.74 |
| 80 | 1 | 0.10 | 1000 | -0.31 | 0.28 | 0.45 | −0.52 | −0.99 | 1.57 | 1.51 | 4.04 | **6.57** | 3.18 | 4.30 |
| 8 | 1 | 0.20 | 1000 | 9.76 | 9.73 | 9.57 | 8.67 | 5.67 | 9.80 | 9.41 | **9.84** | 9.82 | 9.76 | 9.64 |
| 12 | 1 | 0.20 | 1000 | 6.67 | 6.59 | 7.28 | 5.91 | 1.89 | 7.77 | 8.72 | **9.62** | 7.65 | 8.72 | 7.88 |
| 16 | 1 | 0.20 | 1000 | 5.34 | 3.37 | 5.62 | 5.62 | 0.04 | 4.27 | 5.65 | **9.70** | **9.70** | 9.26 | 8.56 |
| 20 | 1 | 0.20 | 1000 | 2.03 | 5.31 | 4.36 | 4.16 | −0.52 | 5.38 | 7.48 | 6.77 | 7.38 | **7.77** | 5.97 |
| 40 | 1 | 0.20 | 1000 | 2.77 | 2.37 | 1.50 | 0.29 | −0.07 | **5.67** | 3.05 | 4.95 | 4.43 | 5.40 | 4.65 |
| 60 | 1 | 0.20 | 1000 | 0.57 | 1.27 | −0.62 | 0.82 | −0.72 | **5.09** | 3.97 | 4.24 | 4.32 | 4.06 | 3.27 |
| 80 | 1 | 0.20 | 1000 | 0.90 | 0.07 | 0.34 | −0.11 | −0.82 | 2.33 | 2.36 | 3.30 | **3.82** | 2.29 | 3.42 |
| 8 | 2+ | 0.00 | 2500 | 4.13 | 6.06 | 6.54 | 3.10 | 2.36 | 8.13 | 8.60 | 9.50 | 8.94 | 8.81 | **9.77** |
| 12 | 2+ | 0.00 | 2500 | 3.07 | 3.21 | 1.04 | 2.33 | −0.13 | 2.21 | 2.45 | 2.72 | 3.28 | 2.70 | **4.26** |
| 16 | 2+ | 0.00 | 2500 | 1.52 | **2.22** | 1.27 | 1.13 | −1.00 | 1.42 | 1.56 | 1.70 | 1.70 | 1.67 | 1.62 |
| 20 | 2+ | 0.00 | 2500 | 1.24 | 1.64 | 1.14 | 1.40 | −0.82 | 1.71 | 1.66 | 1.78 | 1.72 | 1.60 | **1.92** |
| 40 | 2+ | 0.00 | 4000 | 0.40 | 0.56 | 0.52 | 0.26 | −0.57 | 1.32 | 1.52 | **1.80** | 1.46 | 1.70 | 1.51 |
| 60 | 2+ | 0.00 | 4000 | 0.86 | 0.47 | −0.02 | −0.23 | −0.97 | 1.36 | 0.69 | **1.47** | 1.38 | 1.44 | 0.99 |
| 8 | 2+ | 0.10 | 2500 | 3.61 | 6.51 | 7.90 | 3.67 | 2.20 | 6.49 | 8.68 | **9.83** | 8.18 | 8.67 | 8.84 |
| 12 | 2+ | 0.10 | 2500 | 0.82 | 2.02 | 1.32 | 1.16 | 0.60 | 2.65 | 2.56 | 1.79 | 1.77 | **3.32** | 1.84 |
| 16 | 2+ | 0.10 | 2500 | 1.98 | **2.15** | 1.31 | 1.58 | −0.70 | 1.79 | 1.54 | 1.67 | 1.80 | 1.64 | 1.80 |
| 20 | 2+ | 0.10 | 2500 | 1.53 | 1.54 | 0.73 | 1.22 | −0.62 | 1.65 | 1.54 | **1.82** | 1.63 | 1.70 | 1.52 |
| 8 | 2- | 0.00 | 2500 | 3.69 | 5.62 | 3.96 | 2.19 | 3.02 | 5.75 | 7.68 | 7.97 | **9.78** | 9.65 | 8.86 |
| 12 | 2- | 0.00 | 2500 | 0.75 | 0.89 | 0.38 | 0.89 | −0.85 | 0.75 | 0.87 | **2.72** | 0.92 | 2.62 | 1.79 |
| 16 | 2- | 0.00 | 2500 | **1.81** | 0.51 | 0.28 | 0.51 | −0.77 | 0.93 | 0.87 | 0.95 | 0.94 | 0.95 | 0.93 |
| 20 | 2- | 0.00 | 2500 | 0.74 | 0.75 | 0.55 | 0.72 | −1.00 | 0.94 | 0.69 | **0.95** | 0.94 | 0.89 | 0.91 |
| Average | | | | 4.43 | 4.65 | 4.48 | 4.02 | 1.84 | 5.96 | 6.32 | **7.16** | 6.98 | 7.10 | 6.77 |
| Best results | | | | 1 | 3 | 0 | 0 | 0 | 6 | 0 | 19.5 | 9 | 8 | 9.5 |

# C   Extended set of training plots

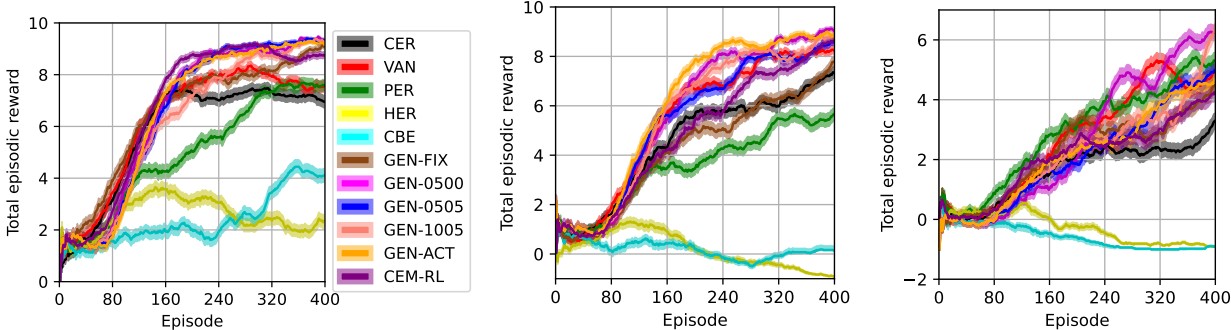

Figure 6: Sample training plots for 1D bit-flipping environment. Each algorithm is run for 10 random seeds, with shaded regions denoting the standard deviation across seeds. The plots correspond to (i) size 6, subgoals 0, (ii) size 7, subgoals 0, and (iii) size 8, subgoals 0.

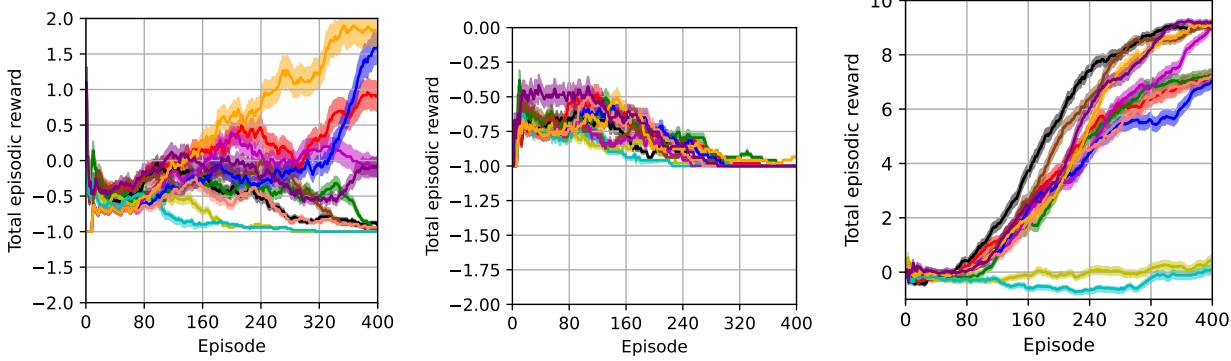

Figure 7: Sample training plots for 1D bit-flipping environment. Each algorithm is run for 10 random seeds, with shaded regions denoting the standard deviation across seeds. The plots correspond to (i) size 9, subgoals 0, (ii) size 10, subgoals 0, and (iii) size 6, subgoals 1.

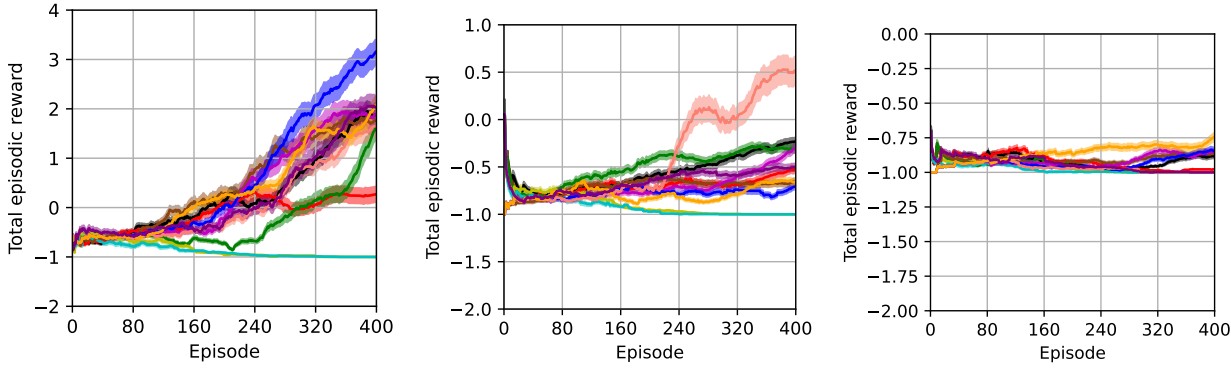

Figure 8: Sample training plots for 1D bit-flipping environment. Each algorithm is run for 10 random seeds, with shaded regions denoting the standard deviation across seeds. The plots correspond to (i) size 7, subgoals 1, (ii) size 8, subgoals 1, and (iii) size 9, subgoals 1.

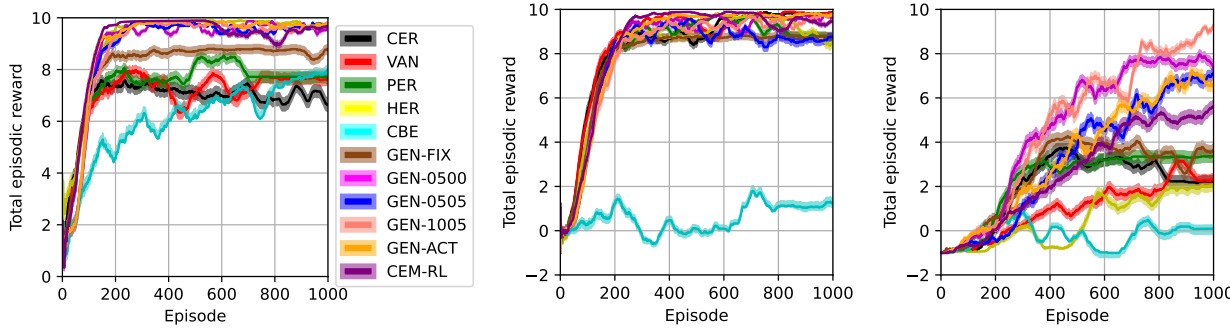

Figure 9: Sample training plots for 2D navigation environment. Each algorithm is run for 10 random seeds, with shaded regions denoting the standard deviation across seeds. Stochasticity is 0, subgoals are 0. The plots correspond to (i) size 8×8, (ii) size 20×20, and (iii) size 80×80.

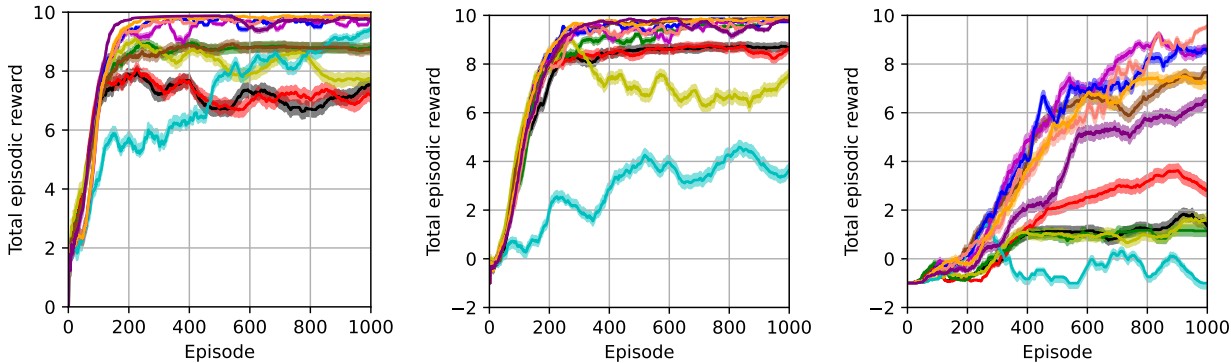

Figure 10: Sample training plots for 2D navigation environment. Each algorithm is run for 10 random seeds, with shaded regions denoting the standard deviation across seeds. Stochasticity is 0.1, subgoals are 0. The plots correspond to (i) size 8×8, (ii) size 20×20, and (iii) size 80×80.

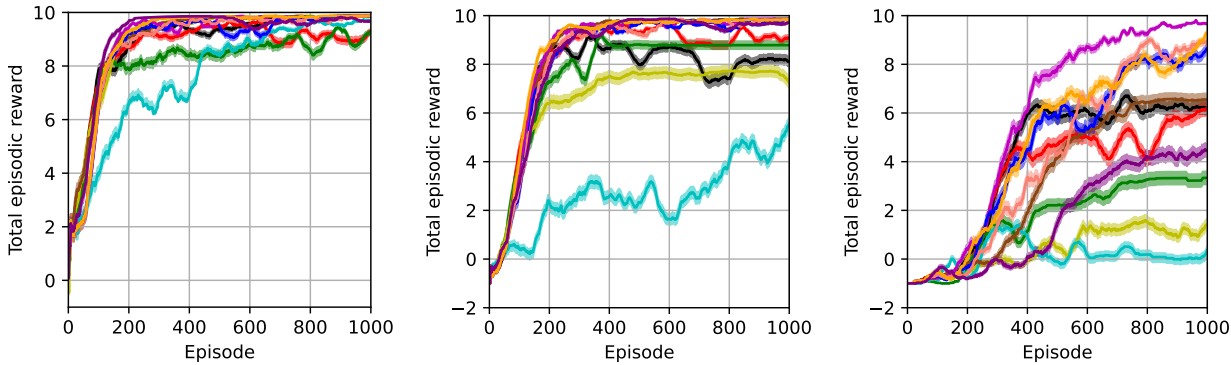

Figure 11: Sample training plots for 2D navigation environment. Each algorithm is run for 10 random seeds, with shaded regions denoting the standard deviation across seeds. Stochasticity is 0.2, subgoals are 0. The plots correspond to (i) size 8×8, (ii) size 20×20, and (iii) size 80×80.

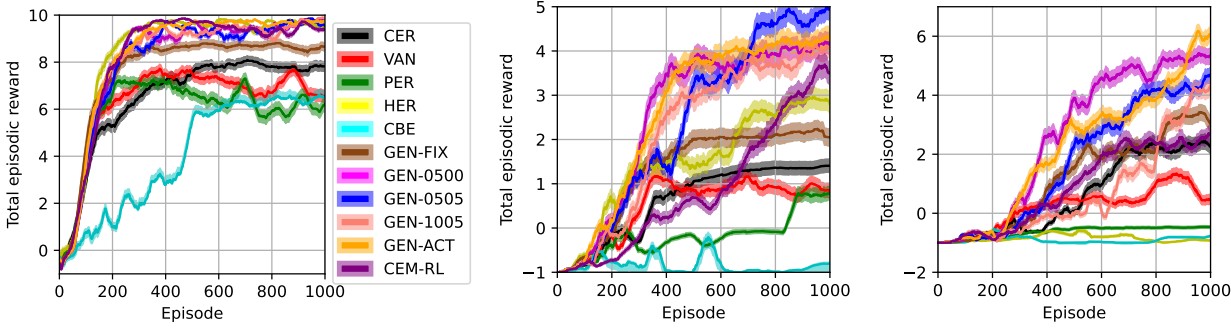

Figure 12: Sample training plots for 2D navigation environment. Each algorithm is run for 10 random seeds, with shaded regions denoting the standard deviation across seeds. Stochasticity is 0, subgoal is 1. The plots correspond to (i) size 8×8, (ii) size 40×40, and (iii) size 80×80.

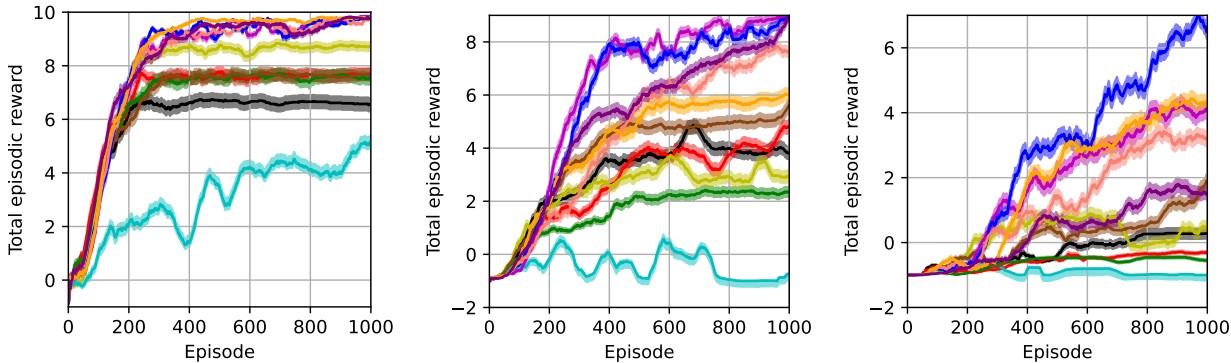

Figure 13: Sample training plots for 2D navigation environment. Each algorithm is run for 10 random seeds, with shaded regions denoting the standard deviation across seeds. Stochasticity is 0.1, subgoals are 1. The plots correspond to (i) size 8×8, (ii) size 20×20, and (iii) size 80×80.

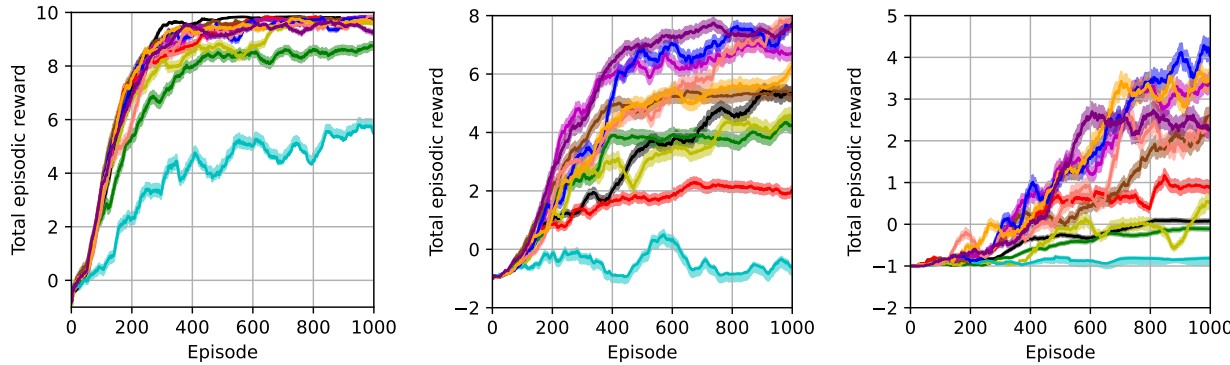

Figure 14: Sample training plots for 2D navigation environment. Each algorithm is run for 10 random seeds, with shaded regions denoting the standard deviation across seeds. Stochasticity is 0.2, subgoals are 1. The plots correspond to (i) size 8×8, (ii) size 20×20, and (iii) size 80×80.

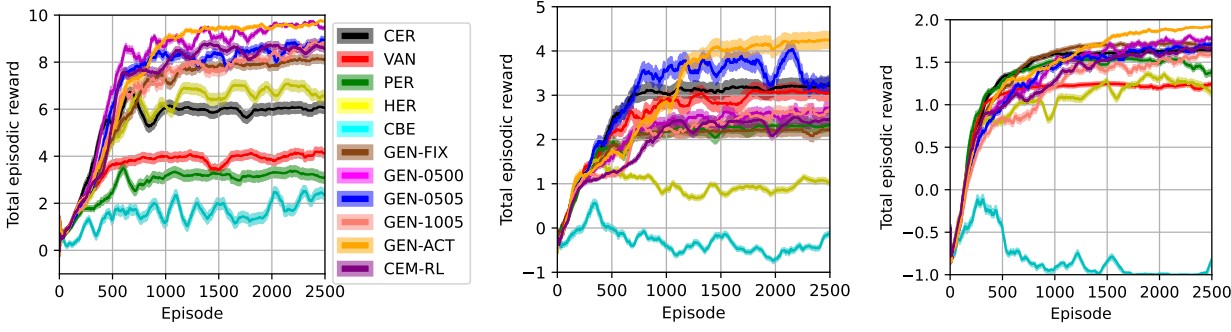

Figure 15: Sample training plots for 2D navigation environment. Each algorithm is run for 10 random seeds, with shaded regions denoting the standard deviation across seeds. Stochasticity is 0, subgoals is 2+. The plots correspond to (i) size 8×8, (ii) size 12×12, and (iii) size 20×20.

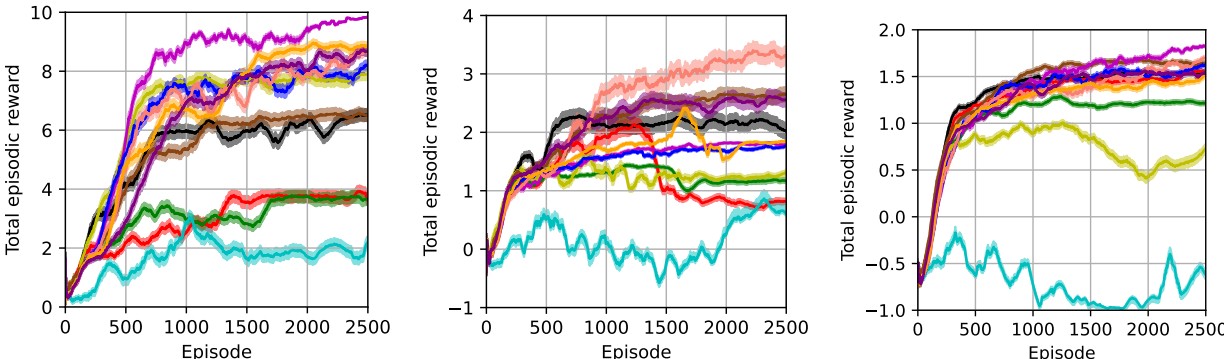

Figure 16: Sample training plots for 2D navigation environment. Each algorithm is run for 10 random seeds, with shaded regions denoting the standard deviation across seeds. Stochasticity is 0.1, subgoals is 2+. The plots correspond to (i) size 8×8, (ii) size 12×12, and (iii) size 20×20.

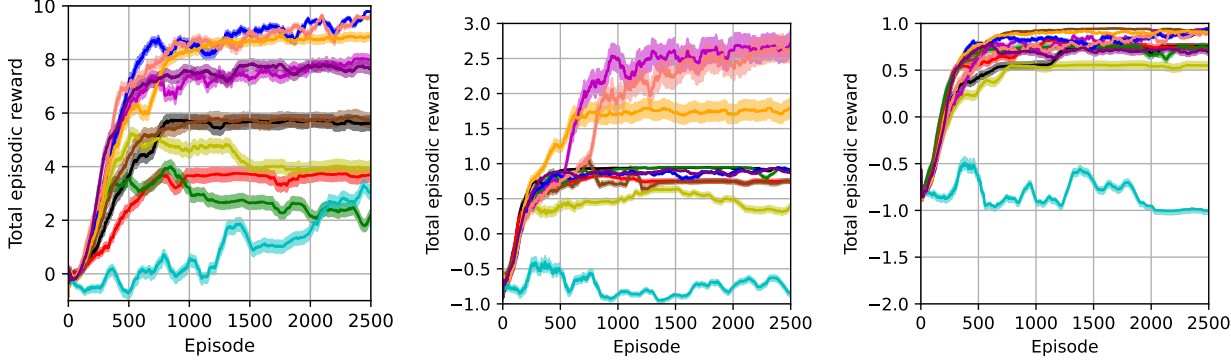

Figure 17: Sample training plots for 2D navigation environment. Each algorithm is run for 10 random seeds, with shaded regions denoting the standard deviation across seeds. Stochasticity is 0, subgoals is 2-. The plots correspond to (i) size 8×8, (ii) size 12×12, and (iii) size 20×20.

