# OpenReview forum: "A Sample Efficient Evolutionary Strategy for Reinforcement Learning"
_TMLR — Rejected by TMLR_

### Review · Reviewer_Sg8r · 2024-01-19

**Summary Of Contributions:**

This article proposes a population-based evolutionary algorithm for improving off-policy reinforcement learning algorithms with a focus on sample efficiency. Instead of evaluating all policies independently and having them learn from independent replay buffers, the authors propose evaluating only a single policy at a time, and updating all policies with a shared experience. This way, the sample complexity of learning is reduced from `population size * policy updates` timesteps (as is typical in population algorithms) to just `policy updates`. However, the number of policy updates (via gradient calculation) remain unchanged. The authors show that their method outperforms several baselines in very simple environments, and hypothesise that this is due to the extra exploration and diversity present in the joint replay buffer.

**Audience:**

Yes

**Claims And Evidence:**

No

**Requested Changes:**

As noted above, there are several analyses that would be needed to support the claims of the article:

1. How does the algorithm scale? To population size, to more complex environments, etc.
2. A systematic protocol for choosing algorithm hyper-parameters, where algorithms are on a level playing field.
3. Analysis of the sampling pattern of policies, and ideally coalescence analysis of ancestry of policies after recombination.
4. Response to reward scale: only positive, only negative, and mixed.
5. Comparison of compute based on gradient updates, not only environment transitions.

**Strengths And Weaknesses:**

## Strengths

The article present a clean idea and is written well. It provides enough details that I have confidence the results are reproducible. The comparison with baselines is adequate, and taking into account recent algorithms. The evaluation is done on very simple environments, but it is sufficiently systematic to show the strengths of the proposed approach.

## Weaknesses

The main figures should be clarified, as it is not obvious from a quick glance, which algorithms are EORL-variants and which are the baselines. I'm assuming the `GEN` prefix is meant to be `EORL`? Also, a better palette would be beneficial: baselines of shades of one colour (e.g. blue) and proposed solutions of another (e.g. red).

It is unclear how well tuned are the hyper-parameters of the baselines. A fairer comparison would be to describe the decision behind them and make sure that all algorithms were subject to the same protocol.

A lot of emphasis is given to covariant matrix adaptation (CMA) algorithms, only to later largely ignore this family of algorithms. I understand that this is an important algorithm in the field of evolutionary strategies, but perhaps its discussion should be sign-posted to indicate that this work does not derive from it? As currently written, I was expecting a CMA-derived algorithm.

The calculation of reward is problematic. Starts at 0 and keeps an exponential moving average with parameter 0.9 (meaning the new data contributes 10% of the aggregated value). Given that agents are sampled at random uniformly, or just the best one, the sampling will be highly dependent on the scale of rewards. For instance, if all the rewards were non-negative, any policy that on average gets non-zero reward would get higher and higher fitness, at least over the first 10-ish episodes. This would lead to a single policy dominating the sampling (except for epsilon deviations). But since the environments in this work are all doling a small negative reward every step, this would encourage sampling different policies at the beginning of the run. I see this as a **major** drawback of the work that needs to be studied and discussed. It would be desirable to present some empirical data on the sampling pattern of policies for running episodes: does one policy dominate? or is there a high turnover?

A population of 8 (and max of 12) is very small for traditional population algorithms where hundreds, and even thousands of agents are the norm. How would we expect this method to scale? My intuition is that this would be disadvantageous to performance because it would highly dilute the sampling, which is compounded by my comment above about fitness calculations.

Another concern I have is that this work is emphasising sample efficiency, but only from the perspective of environment transitions. Every policy still needs to do a gradient update, with the global data. This has two problems: 1) Since other policies are initialised at random, and are performing gradient updates with exactly the same data, they are likely to converge; analysis of how divergent unsampled policies really are would alleviate this concern. 2) Typically the computationally expensive step is a gradient update, as that is typically performed on a GPU or other type of accelerator. Environment transitions are typically performed in CPU and are assumed to be cheap. This is the main reason for algorithms like A3C parallelising the environment transitions, hindering sample complexity (timesteps) but saving GPU (agent updates).

---

### Review · Reviewer_2j6F · 2024-01-19

**Summary Of Contributions:**

This paper presents Evolutionary Operators for Reinforcement Learning (EORL for short), which combines Reinforcement learning with a sparse amount of evolutionary exploration (Random crossover, Linear crossover, and Random mutation), to enhance RL's stability and convergence. EORL maintains a population of RL agents with a shared buffer, and chooses one policy to interact with the environment by  $\epsilon$-greedy method. The empirical evaluation of 1D bit-flipping and 2D grid navigation with subgoals showcase the superiority of EORL, compared to other baseline algorithms.

**Audience:**

Yes

**Claims And Evidence:**

Yes

**Requested Changes:**

1. In 1.2 Related work: "However, their focus is specifically ...... A separate (offline) RL agent learns policies based on this experience."

   Here are some misunderstandings of ERL. The RL agent in ERL also interacts with the environment, so it's not an offline agent.

2. The paper should specify the base RL algorithm of EORL. Is it VAN?

3. The Related work section is kinda outdated and thin, and it will be better if more recent works are added.

4. A more detailed description of the Active Random is needed.

5. More convincing experiments. I would like to see the performance of the proposed algorithm in more complex environments, like OpenAI Mujoco, DMC suite, and so on.

**Strengths And Weaknesses:**

###Strengths:

1. EORL can be a **plug-in** method, which is also easy to use.
2. EORL demonstrates **high performance and parameter robustness** in the experiments section.
3. Thanks to the soft updating fitness, only one policy interacts with the environment at each iteration, which **reduces the interaction overhead**.

&nbsp;

###Weaknesses:

1. **Wrong causal logic** in 1.3 Contributions and usage: "In this paper, we focus on methods and environments that do not require the massively parallel architectures of typical ES methods. We are likely to encounter such constraints wherever parallel simulations are expensive or even unavailable (for example, physical environments or finite element methods). Therefore we assume that only one policy is able to interact with the environment in one episode."

   The statement here is confusing, because it confuses parallel simulation with multiple serial simulations in one iteration. Banning the parallel simulation, you can also evaluating multiple policies in one iteration, which is a common practice in ERL field. And the modifications to CEM-RL out of this consideration are unnecessary.

2. The experiment part includes only two simple environments, on which the empirical result is not credible enough.

---

### Review · Reviewer_wbbz · 2024-02-06

**Summary Of Contributions:**

The paper introduces Evolutionary Operators for Reinforcement Learning (EORL) which combines RL and ES.  The main difference with other approaches such as CEM-RL is that the population is not evaluated at each round but only one policy in the population is evaluated and modified.

**Audience:**

No

**Claims And Evidence:**

No

**Requested Changes:**

As mentioned in the conclusion of the paper, it is crucial to have more experiments on more difficult environments and with continuous actions in order to secure my recommendation.

**Strengths And Weaknesses:**

*Strenghts*: The paper is clear and easy to read.

*Weaknesses*:
1. The experiments are underwhelming as the proposed algorithm was tried on two simple gridworld environments. Evolutionary Operators work on random modifications of the parameters of the neural networks it is then important to test if these modifications have benefits at a larger scale and if these modifications can also work on environments with continuous actions. For example, CEM-RL experimented on MUJOCO.
2. It would be nice to have more insights on the effects of the random perturbations e.g. on the exploration. From a back-of-the-envelope calculation, on the 1D grid world environment, EORL-05-00 only uses the evolutionary operators 10 times on average over the whole training. Would it be nice to see how the evolutionary operators affected the policy.
3. It would be important to have the std on the plots as the algorithms were run on 10 seeds.
4. In Figure 4, sorting the legend with the results would help the understanding as there are lots of curves.
5. What is `gin_fixed` in Figure 5? In addition, the three sub figures from Figure 5 are not aligned.

---

> ### Author Response · Authors · 2024-02-06
> **Response to review wbbz**
>
> We are grateful for the comments, and for the fact that the reviewer found the paper easy to read/understand. Here are our responses to the comments:
> 1. Regarding more experiments: We can understand the reviewer's point of view, and would love to add more experiments to the work. At the moment, we have been unable to do this because (paraphrasing responses to other reviews), (i) we have computational limitations that we cannot fully explain while maintaining anonymity, (ii) we respectfully submit that while the environments are simple in nature, the exploration challenge is still significant because of the size of the state space, and (iii) the purpose of these experiments is to demonstrate the potential of EORL as an additional tool in the box, rather than claim EORL to be a stand-alone SOTA algorithm.
> 2. Regarding the effect of evolutionary updates: The reviewer makes an interesting point. We propose to add heatmaps of the before/after policy q-values to demonstrate this effect, in the revised version.
> 3. Standard deviations: We would be happy to add these numbers. As it stands, Figure 4 and the supplementary figures already contain uncertainty bands marked by shaded regions, which are derived from the standard deviation.
> 4. Figure 4: We can update the legend as suggested.
> 5. Figure 5: Gen_Fixed is the same algorithm as plotted in Figure 4, under the label GEN-FIX. We apologize for the inconsistent naming. We can also fix the alignment of figures in the revised version.

---

### Decision · Action_Editor_3AnL · 2024-03-20

**Recommendation:** Reject

**Comment:**

The paper deals with a relevant problem and contributes potentially interesting ideas, while being for the most part well-written. However, the limited theoretical or empirical study of the proposed algorithm does not fully clarify its value in terms of algorithmic principles for RL. As one reviewer noted, "the experiments are underwhelming and the lack of a deeper analysis on the effect of the evolutionary operators makes it hard to understand if the proposed method brings real improvements."

For these reasons, I am unfortunately recommending rejection.

Nonetheless, I believe that a revised version of the current manuscript that includes substantial theoretical understanding behind the methodology or larger-scale algorithms and experiments would make for a great contribution to the RL community.

**Audience:**

The problem addressed in the paper, i.e., improved algorithms for evolutionary strategies in RL, is relevant for the TMLR community.

**Claims And Evidence:**

The evidence provided by the paper looks too thin to support the claims. With "sample efficiency" mentioned in the title, one may expect some theoretical characterization on the number of interactions taken from the environment. The empirical evidence is also limited to toy domains.

**Resubmission Of Major Revision:**

The authors may consider submitting a major revision at a later time.